# Cephalosporins: A Focus on Side Chains and β-Lactam Cross-Reactivity

**DOI:** 10.3390/pharmacy7030103

**Published:** 2019-07-29

**Authors:** Saira B. Chaudhry, Michael P. Veve, Jamie L. Wagner

**Affiliations:** 1Department of Pharmacy Practice and Administration, Ernest Mario School of Pharmacy, Rutgers University, Piscataway, NJ 08854, USA; 2Department of Infectious Diseases, Jersey Shore University Medical Center, Neptune, NJ 07753, USA; 3Department of Clinical Pharmacy and Translational Science, College of Pharmacy, University of Tennessee Health Science Center, Knoxville, TN 37920, USA; 4Department of Pharmacy Practice, School of Pharmacy, University of Mississippi, Jackson, MS 39216, USA

**Keywords:** cephalosporin, cross-reactivity, side chain, beta-lactam allergy

## Abstract

Cephalosporins are among the most commonly prescribed antibiotic classes due to their wide clinical utility and general tolerability, with approximately 1–3% of the population reporting a cephalosporin allergy. However, clinicians may avoid the use of cephalosporins in patients with reported penicillin allergies despite the low potential for cross-reactivity. The misdiagnosis of β-lactam allergies and misunderstanding of cross-reactivity among β-lactams, including within the cephalosporin class, often leads to use of broader spectrum antibiotics with poor safety and efficacy profiles and represents a serious obstacle for antimicrobial stewardship. Risk factors for cephalosporin allergies are broad and include female sex, advanced age, and a history of another antibiotic or penicillin allergy; however, cephalosporins are readily tolerated even among individuals with true immediate-type allergies to penicillins. Cephalosporin cross-reactivity potential is related to the structural R1 side chain, and clinicians should be cognizant of R1 side chain similarities when prescribing alternate β-lactams in allergic individuals or when new cephalosporins are brought to market. Clinicians should consider the low likelihood of true cephalosporin allergy when clinically indicated. The purpose of this review is to provide an overview of the role of cephalosporins in clinical practice, and to highlight the incidence of, risk factors for, and cross-reactivity of cephalosporins with other antibiotics.

## 1. Introduction

Cephalosporins are a commonly prescribed class of antibiotics in inpatient and community settings due to their clinical utility in a number of infectious disease states. While only 1–3% of the population report a cephalosporin allergy [1,2,3,4], patient-reported or poorly described β-lactam allergies represent a barrier to receiving cephalosporin therapy due to misconceptions related to cross-reactivity. Common clinical practice is to avoid other β-lactam classes, including cephalosporins, in patients with a labeled β-lactam allergy [5]. Additionally, true IgE-mediated allergies are relatively rare and electronic medical record descriptions of β-lactam allergies are often lacking or incomplete [3,5,6,7,8,9]. Most literature suggests that 99% of these patients do not have a true allergy and can safely receive β-lactam antibiotics [6,7].

Previous literature has demonstrated that for general infections, patients with β-lactam allergies are more likely to receive suboptimal therapy, experience clinical failure, have an increased hospital length of stay, develop drug resistant organisms, and have higher in-hospital mortality [10,11,12,13,14]. Since optimal treatment in bacterial infections is often directly associated with improved patient outcomes, and that cephalosporins are often regarded as preferred therapies in many infections, barriers to receiving optimal antibiotics are an important target for antimicrobial stewardship intervention [15].

Among the various cephalosporin generations, each carries different risks for eliciting an allergic reaction, which is primarily driven by the R1 side chain [16,17]. Thus, an allergy to a cephalosporin in one generation may not have any cross-reactivity to another cephalosporin within the same or different generation if the side chains are different [2]. Unfortunately, there are no reliable testing mechanisms to confirm a cephalosporin allergy in a patient [16]. Additionally, desensitization protocols to cephalosporins are not standardized, and test dosing with either an oral or intravenous (IV) administration may not yield accurate results [2]. The objective of this review is to discuss indications where cephalosporin antibiotics are first line therapies, the incidence of cephalosporin allergies, risk factors, and cross-reactivity among cephalosporins and with other β-lactam antibiotics.

## 2. Indications Where Cephalosporins are First Line Therapies

Cephalosporins are a class of antibiotics routinely used for a variety of infections, many of which are recommended first line therapies in North American Infectious Diseases society guidelines such as the Infectious Diseases Society of America (IDSA). In general, there are six different generations of cephalosporins, and drugs in each generation are used in different indications. While the core structure is the same for ß-lactams, changes in position 7 of the β-lactam ring are what differentiates the spectrum of activity for each of the cephalosporin generations [18]. The generations are divided on their order of approval to market and are described in further detail below.

First generation cephalosporins include cefazolin and cephalexin, and these agents are known for their coverage of methicillin-sensitive *Staphylococcus aureus* (MSSA) and streptococci with some Gram-negative bacilli coverage. First generation cephalosporins are commonly prescribed for surgical site infection (SSI) prophylaxis for almost all surgeries, either as monotherapy or as combination therapy [19]. Cefazolin is also used as prophylaxis with the insertion of a cardiac device and as prophylaxis for endometritis [20,21]. This generation is used as first line therapy for Group A streptococcus (GAS) pharyngitis, outpatient treatment for mild diabetic foot infections (DFIs), mild-to-moderate intra-abdominal infections (IAIs), cholecystitis, combat wounds, and prosthetic joint infections (PJIs) [22,23,24,25,26]. Additionally, first generation cephalosporins can be used as alternative therapy for uncomplicated cystitis and alternative prophylaxis against infective endocarditis [27,28]. Cefazolin can also be used as definitive therapy for vertebral osteomyelitis, infections from endoscopic urologic procedures with mucosal trauma, necrotizing fasciitis, pyomyositis, SSIs, and in antibiotic locks [29,30,31,32,33]. More recently, cefazolin, a first-generation cephalosporin, has been examined as a first line agent for treating MSSA infections, including bacteremia and endocarditis [21,34,35,36].

Second generation cephalosporins are broken up into two groups: true second generation cephalosporins and the cephamycins. The true second generation cephalosporins include cefuroxime and cefprozil, while the cephamycins include cefoxitin, cefotetan, and cefmetazole. This class has good coverage against enteric Gram-negative bacilli, *Haemophilus influenzae*, and *Neisseria* spp., with most second generation cephalosporins displaying moderate coverage against streptococcus and staphylococcus. Cefoxitin has moderate coverage of both Gram-positive and Gram-negative anaerobes. Because of the anaerobic coverage, cefoxitin is used prophylactically in multiple surgeries, including cardiac, biliary, appendectomy, small intestine, colorectal, head and neck, hysterectomy, and urologic [19]. Cefoxitin can also be prescribed for use in treating pelvic inflammatory disease (PID), moderate severity DFIs, human and animal bites, early localized or early disseminated Lyme or Lyme-induced arthritis, and mild-to-moderate severity IAIs [23,24,31,37,38]. Alternatively, second generation cephalosporins can be used for treating endometritis, GAS pharyngitis, outpatient treatment of community-acquired pneumonia (CAP), uncomplicated cystitis, and Lyme disease with central nervous system (CNS) involvement but without parenchymal involvement [22,27,39,40,41].

Third generation cephalosporins are the most prescribed cephalosporins and are the first generation to be considered an extended-spectrum cephalosporin. This class includes ceftriaxone, cefotaxime, ceftazidime, ceftazidime/avibactam, cefdinir, cefpodoxime, and cefixime. They are more stable to common β-lactamases produced by Gram-negative bacilli, which offers good coverage against enteric Gram-negative bacilli. However, third generation cephalosporins are hydrolyzed by broad-spectrum β-lactamases, such as extended-spectrum β-lactamases (ESBLs), AmpC-producing organisms, and carbapenemases, among others [42]. Additionally, this class has good coverage against *Streptococcus* spp., moderate coverage against MSSA, and ceftazidime has in vitro activity against *Pseudomonas aeruginosa*. Due to their broad-spectrum of activity, third generation cephalosporins are recommended for a large variety of indications. They are recommended for use as first line prophylaxis against inpatient spontaneous bacterial peritonitis (SBP), biliary or colorectal or liver transplant SSIs, and infections post-urologic procedures, and are recommended as alternative prophylactic agents for neutropenic infections and infective endocarditis [19,28,30,43,44]. This class is heavily used as first line therapy for most infections, including SBP, sexually transmitted infections (gonorrhea, chlamydia, PID, epididymitis, proctitis), moderate-to-severe DFIs, outpatient treatment of CAP, inpatient, non-intensive care unit (ICU) treatment of CAP, outpatient empiric treatment for suspected infection in Human Immunodeficiency Virus (HIV) patients, inpatient empiric treatment for suspected infection in HIV patients, pyelonephritis, necrotizing fasciitis, glanders, SSIs, human and animal bites, Lyme disease with CNS involvement and with or without parenchymal involvement, mild-to-severe IAIs, healthcare-associated (HCA) complicated IAIs, encephalitis, PJIs, HCA meningitis/ventriculitis, and community-acquired meningitis [23,24,26,27,31,37,40,41,43,45,46,47]. Ceftazidime is the only cephalosporin with a Food and Drug Administration (FDA)-approved indication for the inpatient treatment of febrile neutropenia, however, its use is not recommended due to a lack of reliable activity against Gram-negative bacilli and streptococcus [48,49]. Because of ceftazidime’s activity against *P. aeruginosa*, it has garnered use an option to treat hospital-acquired and ventilator-associated pneumonia (HAP/VAP) [50]. Third generation cephalosporins can also be used as alternative treatment options against a variety of infections, including syphilis, gonorrhea, acute bacterial rhinosinusitis, endometritis, GAS pharyngitis, uncomplicated cystitis, infectious diarrhea in Acquired Immune Deficiency Syndrome (AIDS) patients, *Vibrio cholera*, *Yersinia enterocolitica*, combat wounds, PJIs, and community-acquired meningitis [25,26,27,37,45,48,51,52,53,54]. Once susceptibilities for organisms are known, third generation cephalosporins can be used to treat endoscopic urologic procedures with mucosal trauma, vertebral osteomyelitis, skin and soft tissue infections caused by *Nocardia* spp., infectious diarrhea caused by *Salmonella* spp. or *Shigella* spp., infective endocarditis, and antibiotic locks [29,30,31,32,33,36,53,55].

Cefepime is a fourth-generation cephalosporin that has good coverage against MSSA, *Streptococcus* spp., *P. aeruginosa*, and enteric Gram-negative bacilli. Cefepime is commonly used as first line therapy for empiric febrile neutropenia, HAP/VAP, severe DFIs, *P. aeruginosa* isolated in CAP, severe intra-abdominal infections, cholecystitis, cholangitis, HCA biliary infections, PJIs, and HCA meningitis/ventriculitis [23,24,26,40,49,50,56]. It can also be used as alternative therapy for community-acquired meningitis, and as definitive therapy for vertebral osteomyelitis and culture-negative infective endocarditis [29,36,54].

The fifth generation cephalosporins, otherwise known as anti-methicillin-resistant *S. aureus* (MRSA) cephalosporins, include ceftaroline and ceftibiprole. These agents offer good coverage against Gram-positive cocci (e.g., MSSA, MRSA, and *Streptococcus* spp.) and enteric Gram-negative rods, with the exception of extended-spectrum beta-lactamase producers, *Acinetobacter baumanii*, and *Stenotrophomonas maltophilia*. Ceftaroline is available for use in the US and is recommended as a first line agent for treating SSIs [31]. It is listed as an alternative therapy for treating pyomyositis, skin and soft tissue infections in febrile neutropenia, and HAP/VAP [31,50]. Recent data from the CAPTURE trial also suggest that ceftaroline is suitable for treating infective endocarditis [57]. Ceftibiprole offers additional antimicrobial coverage against *Enterococcus faecalis* and *Pseudomonas aeruginosa* and may demonstrate a benefit as alternative therapy in the treatment of nosocomial infections and CAP caused by MRSA [58].

One of the newer cephalosporins, ceftolozane/tazobactam, has not yet been categorized into an existing generation due to its unique spectrum of activity. This agent provides good coverage against enteric Gram-negative bacilli, *P. aeruginosa*, and *Streptococcus* spp. It is currently FDA-approved for treating complicated IAIs and pyelonephritis [59]. Ceftolozane/tazobactam also has results pending for treatment of HAP [60].

## 3. Incidence of Cephalosporin Allergies

The incidence of cephalosporin allergy is estimated to be 1–3% of the general population [1,2,4]. Limitations to allergy documentation in the electronic medical record and falsely patient reported allergies, in addition to difficulties differentiating adverse drug reactions from allergies, can prove difficult to discern the true incidence of cephalosporin allergy [61]. The majority of cephalosporin allergy data is reported in patients with a history of penicillin allergy, but some large claims databases suggest the overall incidence of cephalosporin allergy is low [2]. Macy and Poon performed a cross-sectional analysis on outpatient antibiotic allergy data from 411,543 patients using Kaiser Permanente health-plan from 2007, and found the overall incidence of newly reported cephalosporin allergy within one year after receipt of a cephalosporin prescription to be 1.3% [1]. Similarly, studies from Yoon et al., and Goodman et al. reported the incidence of minor allergic reactions to parental cephalosporins for surgical prophylaxis to be 0.28% and 0.07%, respectively [62,63].

Historically, patients with a penicillin allergy were thought to have contraindications to cephalosporin therapy due to high rates of cross-reactivity [46,64]. However, new more recent data suggests patients with a history of penicillin allergy are roughly 1–4% will have a true cephalosporin allergy [2,3]. Additionally, some experts hypothesize that contamination of pre-1980 cephalosporins with trace amounts of benzylpenicillin may have led to an overestimation of the degree of cross-reactivity between cephalosporins and penicillins [65]. Additional confusion stems from misclassification of an “allergic reaction” to an adverse drug reaction, such as nausea, vomiting, and diarrhea, and penicillin or cephalosporin skin testing is often not performed to confirm allergies [66]. The low rates of cross-reactivity are discussed in a later section of this review.

The most commonly reported cephalosporin allergies include skin manifestations (1–5%), such as maculopapular or morbilliform skin eruption, followed by drug fevers (0.5–0.9%), eosinophilia (2–10%) and anaphylaxis (<0.1%) [2,64]. Regarding serious cephalosporin allergies, Macy and Contreras reported the incidence of anaphylaxis or serious cutaneous adverse reactions to oral or parenteral cephalosporins to be <0.0001% from the electronic claims data from 3.9 million patients and 1.3 million courses of cephalosporin therapy [67]. Additional data suggests cephalosporin-induced anaphylaxis to be 0.0001–0.1%, with differing rates based on the R1 side chain [4,68].

## 4. Risk Factors for Cephalosporin Allergies

Data describing specific risk factors for cephalosporin allergies are limited, and unsurprisingly include a history to penicillin and/or a cephalosporin [2,46]. It is therefore difficult to predict which patients are at high risk of cephalosporin allergy, and a more meaningful approach to cephalosporin allergy risk assessment is to individualize therapy and cross-reaction potential based on patient allergy history.

A key publication from Strom and colleagues established that patients with one antibiotic allergy seemed to be predisposed allergic reactions to other antibiotics, regardless of the potential for cross-reactivity [69]. These findings have also been demonstrated from data focused on β-lactam allergies. A meta-analysis by Pichichero and colleagues, showed a 2.63 increased odds (95% Confidence Interval [CI], 2.11–3.28) of an allergic reaction to any cephalosporin in patients with a reported penicillin or amoxicillin allergy when compared to those without a listed allergy [66]. The investigators also found a 4.8 higher odds (95% CI, 3.7–6.2) of allergic reaction in patients who had received first generation cephalosporins and the second generation cephalosporin cefamandole when compared to other classes of cephalosporins. A potential explanation for these findings could be attributed to similar side chains of penicillin and 1st generation cephalosporins, or that penicillin-allergic patients can display a three-fold increased risk of adverse reactions to any unrelated drugs [66]. Subsequent data suggests individuals with a history of penicillin allergy have a modest increase in odds to have a new cephalosporin allergy report within 30 days of a cephalosporin course (odds ratio [OR], 1.13; 95% CI, 1.07–1.19); however, the overall prevalence of cephalosporin allergy remains low [67]. However, numerous data have suggested that even in confirmed penicillin allergic patients, cephalosporin use is safe due to the low cross-reactivity between different classes of β-lactams [4,46,70].

Macy and Contreras performed a comprehensive review of 1.4 million medical claims data, including patients who received oral and parenteral cephalosporins, and determined that women more frequently reported a new cephalosporin allergy then men (0.56% to 0.43% per cephalosporin course, *p =* <0.0001) [67]. A subsequent cross-sectional analysis from Macy and Poon found older age to be associated with a higher prevalence of cephalosporin allergy in both males (*p* = 0.031) and females (*p* < 0.001) [1]. Additional data describing risk factors for cephalosporin allergy remain an unmet need.

## 5. Cephalosporin Cross-Reactivity

### 5.1. Cephalosporin and Penicillin Cross-Reactivity

Cephalosporins are related to the structure and antimicrobial activity of penicillins. Both groups of antibiotics possess the core four-membered β-lactam ring. The β-lactam ring in penicillins is connected to a five-membered thiazolidine ring, or penam, and the side chain, R, differentiates the different penicillins. In cephalosporins, the β-lactam ring is bonded to a six-membered dihydrothiazine ring, or cepham [46,71]. Both penicillins and cephalosporins are distinct from the other β-lactams, carbapenems and monobactams (See Figure 1).

Up until the mid 1980s, it was believed that the cross-reactivity between early cephalosporins and penicillins was due to cephalosporins being manufactured from the same mold, *Penicillium* spp., as penicillins, which in turn meant due to the shared β-lactam ring [17,71,72,73]. Because of possible contamination from the manufacturing process, first generation cephalosporins were created by modifying the R1 site of the cephalosporin structure. The succeeding generations of cephalosporins have been synthetically produced with modifications at the R1 and R2 sites. These changes in the R site also differentiate the spectrum of activity amongst the cephalosporins [71,73]. The R1 side chain has been found to be the major factor for cross-reactivity to cephalosporins and penicillins; however, some controversy exists surrounding the importance of the R2 side chain in eliciting an immune response [17]. See Table 1 for a summary of β-lactam antibiotics with exact or similar R1 side chains; Table 2 lists β-lactam antibiotics with exact of similar R2 side chains, although R2 side chain similarities are not thought to be pertinent to constituting an immune response. Amoxicillin shares the same side chain as ampicillin, cephalexin, cefadroxil, cefprozil, and cefaclor, cefatrizine. Ampicillin has the same side chain as cefaclor, cephalexin, cephradrine, cephaloglycin and loracarbef. Distinctly, cefazolin does not share a similar side chain with any of the current FDA-approved β-lactams [46,74].

There are mixed literature in regards to the extent of cross-reactivity between penicillins and cephalosporins. Older studies and case reports before 1980 claimed the cross-reactivity between benzyl-penicillin and first and early second generation cephalosporins to be up to 10% and 2–3% in third generation cephalosporins [75,76]. Currently, the US Food and Drug Administration adapted this old data and labeled cephalosporins to potentially have a 10% cross-reactivity [77]. Newer data has claimed there to be a 2–5% reactivity between penicillins and cephalosporins, based on 12 post 1980s studies, which included 417 patients and considered the positive predictive value of the penicillin skin testing to be 40% to 100% [1,77,78]. Because drug allergic patients can develop allergic reactions by non-cross-reacting compounds, these reactions to cephalosporins in penicillin allergic patients may not truly reflect cross-reactivity between the two classes [74].

The cross-reactivity between first generation cephalosporins and penicillins was examined in a meta-analysis that compared allergic reactions to a cephalosporin in a penicillin allergic and non-penicillin allergic patients [66]. This study included a total of nine articles, in which it was determined that there was a significant increase in allergic reactions to cephalothin (OR: 2.5; 95% confidence interval (CI): 1.1–5.5), cephaloridine (OR: 8.7; 95% CI: 5.9–12.8), cephalexin (OR: 5.8; 95% CI: 3.6–9.2), and all other first generation cephalosporins and cefamandole were found to have penicillin allergic reactions (OR: 4.8, 95% CI: 3.7–6.2). There was not an increased risk with the second generation cephalosporins (OR:1.1; 95% CI: 0.6–2.1) or third generation cephalosporins (OR: 0.5: 95% CI: 0.2–1.1). This study determined that first generation cephalosporins and penicillins have cross allergenicity, whereas there is negligible risk with second and third generation cephalosporins. This study also helped validate that cross-reactivity to the first generation cephalosporins could be due to the similar R1 side chains that they share [66]. Due to the amino-penicillins sharing a similar R1 side chain to many first and second generation cephalosporins, another study evaluated the cross-reactivity between amoxicillin and cefadroxil and cefamandole (which has a different side chain than amoxicillin). This study demonstrated a strong correlation to cross-reactivity between penicillins and similar side chains with 8/21 (38%) with an amoxicillin allergy and having a positive response to cefadroxil and no patients reacted to cefamandole [79]. Therefore, it is recommended to do pretreatment skin tests with cephalosporins with similar side chains in penicillin allergic patients [76].

Cross-reactivity to penicillins and cephalosporins with different side chains have been infrequently reported. One study looked at 34 penicillin allergic patients and tested them for skin test reactivity to amoxicillin, cephalexin, and ceftazidime. It was found that only 5 patients (14%) had reactivity to cephalexin and none to ceftazidime [80]. In a study of 128 penicillin allergic patients, 14 (10.9%) patients had positive skin test reaction to cephalosporins, of which 9 patients had positive skin tests to cephalothin and cefamandole. When the cefamandole and cephalothin positive skin test patients and negative cephalosporin skin test patients were challenged to receive ceftriaxone or cefuroxime, none of the 101 patients had a reaction [81]. It was demonstrated in one study that 17 out of 19 (89.4%) patients tolerated therapeutic doses of cephaloridine and cefamandole [82]. A similar type study found in 41 penicillin allergic patients that all tolerated therapeutic doses of cefazolin, cefuroxime, and ceftriaxone, all of which had different side chains than the penicillin responsible for the reaction [83]. Of note, it seems that cefazolin has selective hypersensitivity and does not have a lot of cross-reactivity with penicillins. As seen in the study by Uyttebroek and et al., where 16 out of 19 with IgE-mediated hypersensitivity to cefazolin, were able to tolerate challenges with either amoxicillin, amoxicillin-clavulanic acid or cefuroxime axetil [84].

There have been many observational studies where cephalosporins were given to patients with penicillin allergies. These studies did not specifically depend on skin testing to determine penicillin allergies. Goodman and et al. safely gave 300 out of 413 (73%) penicillin allergic patients with a planned orthopedic surgery a cephalosporin. All patients received cefazolin except one who received ceftazidime. Only one patient had an allergic reaction (0.3%) [63]. In an intraoperative antibiotic study looking at patients with reported penicillin allergies and who were given a cephalosporin, none of the 6067 patients were reported to have an allergic reaction [85]. In another peri-operative antibiotic study in penicillin allergic patients, a total of 513 allergic patients were identified, encompassing 624 surgical patients. In these patients, 153 were given a cephalosporin, (with cefazolin used 83% of the time), and only one patient experienced anaphylaxis [86]. In a study looking at 420 peri-operative patients, 147 patients stated a penicillin allergy, of which 84 patients received a cephalosporin. None of the 84 patients had a reaction [87]. About 270,155 patients were looked at within in one healthcare plan, with a penicillin allergy and were given a course of a cephalosporin. This study found that a greater proportion of patients with a penicillin history (1.13%; 95% CI, 1.07–1.19%) had a new report of a cephalosporin allergy within 30 days of a cephalosporin course than those with no reports of a drug allergy (0.39%; 95% CI, 0.37–0.40%) [67]. Jeffres et al. observed in 13 patients with Gram-negative bacteremia and a history of a penicillin allergy, that seven of these patients reacted to a cephalosporin administered to them [14]. Lastly, Crotty and et al. reviewed 175 patients with self-reported penicillin allergies in a hospital setting who received one of the following cephalosporins, cefepime, cefoxitin, ceftriaxone, and cephalexin. Allergic reactions were only observed with cefepime (6 out of 96 patients) and cefoxitin (1 out of 13 patients). Thus, these studies demonstrated that the incidence of penicillin allergy cross-reactivity is low, especially with those cephalosporins with different side chains [88]. Therefore, many of these studies concluded that it is safe to administer cephalosporins in patients with a history of penicillin allergy.

### 5.2. Cross-Reactivity among Cephalosporins

Like penicillins, cephalosporins can cause IgE-mediated reactions, which usually present as urticaria, angioedema, rhinitis, bronchospasm, and anaphylactic shock. These symptoms can occur within 1 h after administration [89]. Literature has proven that side chains are responsible for the cross-reactivity amongst the cephalosporins, mainly at the R1 side chain site. This is due to the structure of the cephalosporin-hapten complex. During the cephalosporin degradation process, loss of the R2 group occurs by rupturing the dihydrothiazine ring and the R1 group remains intact [17,74]. Many cephalosporins share the same moiety at the R1 site. The R2 side chain, however, may have a role in immunogenicity, but to a lesser extent as seen in in-vitro and clinical studies.

Moiety changes within the side chains may also contribute to cross-reactivity between the cephalosporins. This was confirmed by Antunez et al., in 24 patients with an IgE-mediated hypersensitivity to cephalosporins who underwent skin tests and RAST (Radioallergosorbent Assay Technique) with a panel of penicillins and cephalosporins. In this study they also performed RAST-inhibition studies to establish cross-reactivity. Twenty-one patients had positive skin tests to cephalosporins, with twelve patients specifically to the culprit cephalosporin and nine patients to more than 1 cephalosporin. Of the nine patients, five were positive to cefuroxime, cefotaxime, and ceftriaxone, two to cefuroxime and cefotaxime, and one to ceftriaxone, cefotaxime, and ceftazidime [90]. Cefuroxime, ceftriaxone, cefotaxime, and cefepime each have a methoxyimino group in the R1 side chain and cross-reactivity between these cephalosporins has been noted [91]. Uniquely, ceftazidime has a R1 side chain different than ceftriaxone, cefepime, and cefuroxime. Ceftazidime’s R1 side chain has an alkoxyimino group that has greater steric hindrance than the methoxyimino group, therefore, it would not be acknowledged by the same IgE molecules [74].

Romano and et al. studied cross-reactivity and tolerability of alternative cephalosporins in 102 patients with IgE-mediated hypersensitivity to cephalosporins via skin tests and serum specific IgE assays and challenges. Skin tests were performed with benzylpenicillin, reagents, ampicillin and amoxicillin, and eleven different cephalosporins (i.e., cephalexin, cefaclor, cefadroxil, cefazolin, cefamandole, cefuroxime, ceftazidime, ceftriaxone, cefotaxime, cefepime, and ceftibuten), and any other responsible cephalosporins. Subjects were categorized into four groups: group A, positive responses to one or more of ceftriaxone, cefuroxime, cefotaxime, cefepime, cefodizime, and ceftazidime; group B, positive responses to amino-cephalosporins (i.e., cefaclor and cephalexin); group C, positive response to cephalosporins other than those belonging to the aforementioned groups; and group D, positive response to cephalosporins belonging to two different groups. Group A had an *n* = 73, group B had an *n* = 13, group C was an *n* = 7, and group D had an *n* = 9. In group A, 41 patients were positive only to their culprit cephalosporin (which was mainly ceftriaxone). Whereas 32 had different cross-reactivity patterns. In group B, 11 patients displayed positive response only to their responsible cephalosporin (9 to cefaclor and 2 to cephalexin), and two had cross-reactivity patterns. In groups A and B, it was demonstrated that the cross-reactivity was due in part to the similar R1 side chains. In group C, 6 were positive only to the responsible agent (5 to cefazolin and cefamandole (*n* = 1)) and one reacted to cefoperazone and was positive to both cefoperazone and cefamandole. These two cephalosporins share identical R2 side chains. Lastly, the nine subjects in group D exhibited different patterns of positivity, which may be due to co-existing sensitivities and not by similar or identical side chains. Challenges with alternative cephalosporins were well tolerated. This study concluded that cephalosporin hypersensitivity is not a class hypersensitivity [89].

Another study reported that twenty (83.3%) of twenty-four cephalosporin allergic patients were allergic only to their index cephalosporin by specific IgE (sIgE) testing to penicillin, amoxicillin and cefaclor, followed by skin prick testing (SPT), intradermal testing (IDT), and drug provocation testing (DPT) with a panel of penicillins and cephalosporins. One ceftriaxone allergic patient had a positive IDT to cephalexin. A cefazolin allergic patient, confirmed on DPT, was allergic to cephalexin on DPT. These are not explainable by side chain similarities and these patients were not sensitized to other cephalosporins or penicillins; therefore, the authors suggest that these were examples of co-sensitization rather than cross-reactivity by way of class effect [3]. 

### 5.3. Cephalosporin Cross-Reactivity with Monobactams and Carbapenems

Monobactams and carbapenems are two other types of β-lactam antibiotics. Monobactams have a monocyclic ring structure and the carbapenems have a bicyclic nucleus comprised of a β-lactam ring with an accompanying five-membered ring [89]. Although monobactams and carbapenems are widely used, studies are lacking in evaluating the cross-reactivity of aztreonam and carbapenems in patients with IgE-mediated hypersensitivity to cephalosporins. With the exception of one study that evaluated 98 patients with an IgE-mediated hypersensitivity to cephalosporins (mainly ceftriaxone, ceftazidime, and cefotaxime) by skin tests and serum specific IgE assays with penicillins, and skin tests with aztreonam, imipenem/cilastatin, and meropenem. One patient had a positive result to aztreonam, and one patient had two anaphylactic reactions to both aztreonam and ceftazidime (which share identical side chains). One patient was also positive to both meropenem and imipenem/cilastatin and all the other reagents tested [89]. Contrary to these findings, Moss and et al. did show tolerability of aztreonam in four patients with cystic fibrosis allergic to ceftazidime [92]. A meta-analysis of all studies in adults and children reviewed the reactions to the carbapenem in patients with IgE hypersensitivity to penicillins and cephalosporins. The reactions were classified as proven, suspected, or possible IgE-mediated and non-IgE-mediated. A total of twelve patients were identified to have cephalosporin hypersensitivities and the incidence of any type of hypersensitivity reaction to a carbapenem was 3/12 (25%); which included two non-IgE-mediated reactions (with imipenem and meropenem) and one possible IgE-mediated reaction to imipenem [93]. Of note, a case report did describe a patient developing delayed type hypersensitivity to ceftriaxone and meropenem, which was more likely due to non-immediate T-cell mechanisms and the diagnosis workup could not establish clear cross-reactivity between the two drugs [94]. Overall, the literature suggests that there is minimal cross-reactivity between cephalosporins and monobactams and carbapenems, however, there is limited data in investigating these cross-reactivities.

## 6. Conclusions and Implications for Antimicrobial Stewardship Programs

The overall prevalence of true cephalosporin allergy is extremely rare, and is likely potentiated by inappropriate, patient-reported β-lactam allergies. Most patients with a penicillin allergy can safely receive a cephalosporin, and many cephalosporins do not cross-react with each other. The conundrum of cephalosporin allergies can be mitigated by an active antimicrobial stewardship program. Improved allergy documentation and clarification practices in the electronic medical record, such as requiring a documented allergy reaction and standardized clinical history regarding cephalosporin allergy, can help clinicians to triage clinically significant β-lactam allergies in order to avoid the use of a non-cephalosporin in appropriate situations. In patients with true IgE-mediated β-lactam reactions, clinicians should avoid prescribing agents with exact or similar R1 side chains as assessed through chemical structures.

## Figures and Tables

**Figure 1 pharmacy-07-00103-f001:**
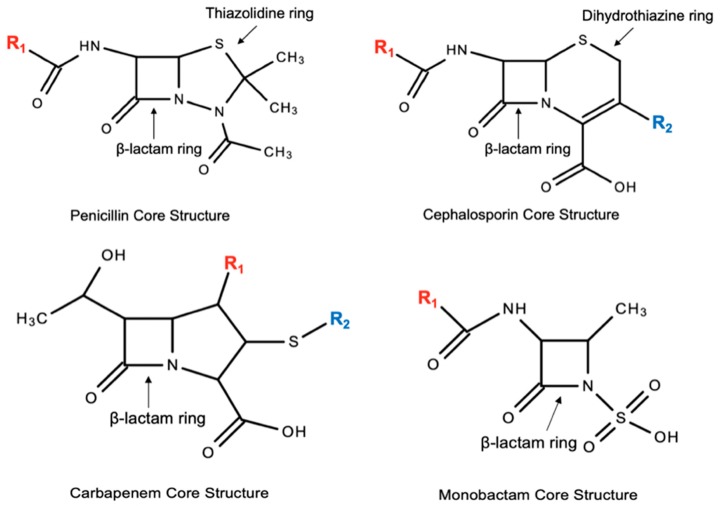
Structural similarities between penicillins, cephalosporins, carbapenems, monobactams, and common β-lactam ring. R1 side chains are depicted in red; R2 side chains are depicted in blue.

**Table 1 pharmacy-07-00103-t001:** β-lactam antibiotics with exact or similar R1 side chains [17,46].

β-Lactam	Agents with Exact or Similar R1 Side Chains	
	Exact R1 Side Chains	Similar R1 Side Chains
Cefaclor	Ampicillin, Cephalexin	Amoxicillin, Cefadroxil, Cefatrizine, Cefmandole, Ceefonicid, Cefprozil, Penicillin G, Penicillin V
Cefadroxil	Amoxicillin, Cefatrizine, Cefprozil	Ampicillin, Cefaclor, Cefmandole, Cefonicid, Cephalexin, Penicillin G, PenicillinV, Piperacillin
Cefatrizine	Amoxicillin, Cefadroxil, Cefprozil	Ampicillin, Cefaclor, Cefmandole, Cefonicid, Cephalexin
Cefazolin	Ceftezole	
Cefditoren	Cefepime, Cefodizime, Cefotaxime, Cefpirome, Cefpodoxime, Ceftriaxone	Ceftaroline, Ceftolozane
Cefepime	Cefditoren, Cefotaxime, Cefozidime, Cefpirome, Cefpodoxime, Ceftriaxone	Ceftaroline, Ceftolozane
Cefiderocol	Aztreonam, Ceftazidime	Ceftaroline
Cefixime		Ceftaroline, Ceftolozane
Cefmandole	Cefonicid, Cefprozil	Amoxicillin, Ampicillin, Cefaclor, Cefadroxil, Cefatrizine, Cephalexin, Penicillin G, Penicillin V, Piperacillin
Cefodizime	Cefditoren, Cefepime, Cefotaxime, Cefpirome, Cefpodoxime, Ceftriaxone	Ceftaroline, Ceftolozane
Cefonicid	Cefmandole	Amoxicillin, Ampicillin, Cefaclor, Cefadroxil, Cefatrizine, Cefprozil, Cephalexin, Penicillin G, Penicillin V, Piperacillin
Cefotaxime	Cefditoren, Cefepime, Cefodizime, Cefpirome, Cefpodoxime, Ceftriaxone	Ceftaroline, Ceftolozane
Cefoxitin	Cephalothin	Cephalothin
Cefpirome	Cefditoren, Cefepime, Cefodizime, Cefotaxime, Cefpodoxime, Ceftriaxone	Ceftaroline
Cefpodoxime	Cefditoren, Cefepime, Cefodizime, Cefotaxime, Cefpirome, Ceftriazone	Ceftaroline, Ceftolozane
Cefprozil	Amoxicillin, Cefadroxil, Cefatrizine	Ampicillin, Cefamandole, Cefonacid, Cephalexin
Ceftaroline		Cefditoren, Cefepime, Cefodizime, Cefotaxime, Cefpirome, Cefpodoxime, Ceftazidime, Ceftriaxone
Ceftazidime	Aztreonam, Cefiderocol	Ceftaroline
Ceftolozane		Cefpodoxime, Ceftriaxone, Cefixime, Cefotaxime, Cefodizime, Cefepime, Cefditoren
Ceftezole	Cefazolin	
Ceftriaxone	Cefditoren, Cefepime, Cefodizime, Cefotaxime, Cefpirome, Cefpodoxime	Ceftaroline, Ceftolozane
Cephalexin	Ampicillin, Cefaclor	Amoxicillin, Cefadroxil, Cefamandole, Cefatrizine, Cefonicid, Cefprozil, Penicillin G, Penicillin V
Cephalothin	Cefoxitin	

**Table 2 pharmacy-07-00103-t002:** β-lactam antibiotics with exact or similar R2 side chains ^†^ [17,46].

β-Lactam	Agents with Exact or Similar R2 Side Chains	
	Exact R2 Side Chain	Similar R2 Side Chain
Cefazolin		Ceftezole
Cefdinir	Cefixime	
Cefepime		Cefiderocol
Cefiderocol		Cefepime
Cefixime	Cefdnir	
Cefmandole	Cefoperazone, Cefotetan	Cefonici
Cefonicid		Cefmandole, Cefoperazone, Cefotetan
Cefotaxime	Cephalothin, Cephapirin	Cefuroxime
Cefoxitin	Cefuroxime	Cefotaxime, Cefoxitin, Cephapirin
Cefoperazone	Cefamandole, Cefoperazone	Cefonicid
Cefpirome		Ceftazidime
Cefotetan	Cefmandole, Cefoperazone	Cefonicid
Ceftezole		Cefazolin
Cefuroxime	Cefoxitin	Cefotaxime, Cephalothin, Cephapirin
Cephalothin	Cephapirin	Cefoxitin, Cefuroxime
Cephapirin	Cephalothin	Cefoxitin, Cefuroxime

^†^ The R2 side chain cross-reactivity is thought to play a less significant role than R1 side chain cross-reactivity in constituting an immune response.

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
