# Peer review of "Cephalosporins: A Focus on Side Chains and β-Lactam Cross-Reactivity"

_pharmacy, 2019, doi:10.3390/pharmacy7030103_

Round 1
Reviewer 1 Report
Dear authors,
thank you for this detailed overview over Cephalosporins and cross-reactivity depending on the side-chain structure.
Nevertheless the manuscript is well-structured and very detailed, I have some questions and recommendations to improve the readability.
In my point of view it would be an improvement to include Figure 1 already earlier in the manuscript. Maybe directly behind paragraph 1 with a short explanation or paragraph of the general structure. Especially for readers not completely familiar with these structures this could be very helpful.
In part two you describe the certain generations of Cephalosporins and their field of application but I miss information about their differences. E.g Line 78: true second generation cephalosporins and cephamycins. It could improve readability to include a short explanation whats the structural difference leading to this distinction. Maybe a figure with a general strucutre could be helpful.
Even if its common knowledge please provide the correct wording for each abbreviation
Table 1. In my point of view confusing. It is not directly obvious if and which side chains are exact or similar. I miss information if cross-reactivities are exsisting or not.
Same problem like in table 1. There is no information about the structure of a side-chain. Are there side-chains R1 which "normally" cause cross-reactivities and others which do not? When describing different generation Cephalosporins do they have different side-chains?
Please provide the structure of monobactams and carbapenems
Reviewer 2 Report
Review, pharmacy-547934. Cephalosporins: A Focus on Side Chains and β-Lactam Cross-Reactivity
The manuscript (MS) addresses allergy to cephalosporins and cross-reactivity to other beta-lactams with a particular focus on the R1 side chain. The MS includes a competent description of the five major cephalosporin “generations” and lists infections where the use of the different classes is indicated. The penicillin and cephalosporin core structures are depicted, and a Table compares agents with exact or similar side chains. The MS is relatively short but comprehensive regarding the topic, very well written, and includes 92 references.
Cephalosporins continue to be a very important class of antimicrobials, and cephalosporin allergy and cross-reactivity are frequently referred to. A PubMed search of cephalosporin allergy with the “Review” filter activated returns 276 search results; however, no recent review covers this particular focus.
Specific points.
The selected audience is clinicians, and the MS makes reference to guidelines. Guidelines are not universal, and referred guidelines (North-American?) should be clearly stated or briefly mentioned in the text. It is unexpected that second generation cephalosporins have “moderate coverage against streptococcus and staphylococcus” (Line 82), because cefuroxime is widely used for treatment of MSSA including meningitis; also, the authors state (L. 89) that second generation cephalosporins can be used for treating GAS pharyngitis.
L. 95. Third generation cephalosporins "are more stable to common β-lactamases produced by Gram-negative bacilli"; the statement need to be clarified, since species-specific β-lactamases of Citrobacter, Enterobacter, Serratia and several other Enterobacterales are capable of degrading 3rd generation cephalosporins.
Reviewer 3 Report
Saira B. Chaudhry and colleagues have written this review about Cephalosporins and its cross reactivity. it is a very good and thorough review paper discussing the Cephalosporins family of compounds and covers all aspects of Cephalosporins usage and its five different generation. This review provides very good and detailed information about Cephalosporin related Allergies and cross reactivities. The selection and number of references is suitable for a review paper in this size. I recommend it for publication in its current form.
Author Response
Reviewer 3 did not have comments or recommendations needed for the manuscript.
Thank you!
Saira Chaudhry
Round 2
Reviewer 1 Report
Dear authors,
thank you for your kind reply and exlpanation.
But I still have two small suggestions/recommendations which could further improve the understandin gof your really interesting and helpful manuscript.
For table 1 and 2 I would suggest instead of the superscript a and b two separate columns. One for exact side chain and one for similar side chains.
Also especially in table 2 are a lot of doubles (e.g line 3 and 4 Cefepime and Cefiderocol). Maybe you can eliminate those.
Also I wondered if the drugs mentioned in table 2 have the same/similar R1 side chain or not. As I found there drugs mentioned not part of table 1 think not. Maybe one sentence to clarify this (in the table description or text) could be useful too.
